# Non-Coding RNAs and Gut Microbiota in the Pathogenesis of Cardiac Arrhythmias: The Latest Update

**DOI:** 10.3390/genes14091736

**Published:** 2023-08-30

**Authors:** Naoko Suga, Yuka Ikeda, Sayuri Yoshikawa, Kurumi Taniguchi, Haruka Sawamura, Satoru Matsuda

**Affiliations:** Department of Food Science and Nutrition, Nara Women’s University, Kita-Uoya Nishimachi, Nara 630-8506, Japan; fyb83bh720fj@gmail.com (N.S.); tyvufkxaq1226-218@outlook.jp (Y.I.); lilyluckriver12@gmail.com (S.Y.); paris.walnut1248.2@icloud.com (K.T.); swmuuu55@icloud.com (H.S.)

**Keywords:** ncRNA, lncRNA, miRNA, cardiac arrhythmia, atrial fibrillation, gut microbiota, APRO family protein

## Abstract

Non-coding RNAs (ncRNAs) are indispensable for adjusting gene expression and genetic programming throughout development and for health as well as cardiovascular diseases. Cardiac arrhythmia is a frequent cardiovascular disease that has a complex pathology. Recent studies have shown that ncRNAs are also associated with cardiac arrhythmias. Many non-coding RNAs and/or genomes have been reported as genetic background for cardiac arrhythmias. In general, arrhythmias may be affected by several functional and structural changes in the myocardium of the heart. Therefore, ncRNAs might be indispensable regulators of gene expression in cardiomyocytes, which could play a dynamic role in regulating the stability of cardiac conduction and/or in the remodeling process. Although it remains almost unclear how ncRNAs regulate the expression of molecules for controlling cardiac conduction and/or the remodeling process, the gut microbiota and immune system within the intricate networks might be involved in the regulatory mechanisms. This study would discuss them and provide a research basis for ncRNA modulation, which might support the development of emerging innovative therapies against cardiac arrhythmias.

## 1. Introduction

Cardiac arrhythmias are a group of heterogeneous disorders in the heartbeat, which may be often defined as any variations in the rate of the normal heart. Abnormal impulse formations and disturbances in conduction may be two major reasons for arrhythmias. However, the physiological dysfunctions linked to the specific arrhythmias are very intricate. The primary pace-making activity of the heart is determined by a spontaneous action potential within atrial node cells [1], which generates cardiac excitation. At present, it has been suggested that the distinctive pace generation might characteristically rely on a probable membrane clock mechanism. The membrane clock consists of plasma membrane ion channels carrying an inward Na^+^/Ca^2+^ exchange current, which might stimulate the surface sarcolemmal electrogenic membrane clock [2]. Therefore, the rhythm-associated ion channels related to the instigation or progression of the action potential have been well recognized. Almost certainly, arrhythmias are mostly caused by imbalances of these calcium ion channels and dysregulations of conduction in heart muscles. The identification of genetic components underlying cardiac arrhythmias has also highlighted the specific role of ion channels. Most ion channels are typically protein complexes that are positioned in the cardiomyocyte sarcolemma, and they might play a role in calcium ion flow conduction [3]. Shifts in the balance of these currents could either increase or decrease the duration of the action potential. Inhomogeneous polarization, influencing the heart, may potentially lead to arrhythmias [3].

Non-coding RNAs (ncRNAs) have been revealed to regulate a diversity of ion channels and/or intercellular linking proteins/molecules, suggesting that ncRNAs may be an important regulator of cardiovascular diseases, including arrhythmias [4]. In fact, it has been revealed that ncRNA could regulate the development of various cardiac diseases [5]. Therefore, recent findings on the implication of ncRNAs in the development of the most common forms of cardiac arrhythmias with a focus on therapeutic and/or clinical application have been discussed here. The ncRNAs are around 200-nt base sequences that might regulate genetic and epigenetic gene expression as well as intracellular signaling mechanisms [6]. In general, ncRNAs have been used as biomarkers for diagnosis and/or treatment due to their involvement in the instigation of diseases. Hence, ncRNA has also been investigated in the field of cardiovascular diseases [7] (Figure 1). The ncRNAs have enormous importance in clinical applications and are usually classified into several categories, such as miRNAs, circRNAs, and/or lncRNAs [8]. For example, recent studies have shown that the association between lncRNAs and proteins may be related to cardiac arrhythmias. Short interference RNA (siRNA) could be often used to target specific ncRNAs. RNAi-mediated siRNAs are highly adjustable and could be used to suppress their mRNA’s encoding protein. The miRNAs are approximately 22-nucleotide RNA molecules that could regulate cell signaling and/or decrease the expression of specific genes by modifying the translation procedure. These ncRNAs have emerged as significant components of transcriptional regulatory pathways that could manage various signaling pathways, cardiac development, stress response, and remodeling in cardiac pathology [9].

Although more than a few arrhythmogenic mechanisms, such as alterations in the function of calcium ion channels and/or excessive oxidative stresses in cardiac cells, have been widely studied, there are still some vacancies to be clarified. To a certain extent, the understanding of pathogenesis for the prevention and/or treatment of cardiac arrhythmias without any side effects has become an important issue to be solved urgently. In this regard, the gut microbiota might have a significant impact on cardiac arrhythmias through a variety of mechanisms, suggesting a broad range of conceivable treatment options without many side effects [10]. In addition, a related study suggests that cardiac arrhythmias may be associated with nutrient intake and/or some metabolisms [11]. Interestingly, convincing evidence exists for weight loss and the management of associated diets to improve outcomes of the treatment [11]. The exploration of the connected mechanisms of dysbiosis of the gut microbiota, including the increase of harmful metabolites, has been gradually discovered. On this basis, it has been revealed that abnormal metabolites of gut dysbiosis might be linked to cardiac arrhythmias [12]. Now, comprehensive studies related to the arrhythmogenicity of ncRNAs have been briefly summarized, which might provide a theoretical basis for novel treatment strategies inspiring the prevention and/or treatment of cardiac arrhythmias.

## 2. Atrial Fibrillation

Atrial fibrillation is an asymmetrical heart rhythm that might commonly lead to heart palpitations. Interestingly, individuals carrying the non-coding 4q25 locus near the *PITX2* gene were 60% more susceptible to this abnormal heart rhythm [13]. Similarly, lncRNAs may be potentially associated with the development of atrial fibrillation. Several lncRNAs, including *LINC00844*, *RP11-532N4.2*, *UNC5B-AS1*, *RP3-332B22.1*, *RP11-432J24.5*, and *RP11-557H15.4,* have been shown to be differentially expressed in patients with atrial fibrillation compared to normal individuals, whose target might be related to the calcium signaling pathway and/or toll-like receptor signaling pathway [14,15]. In addition, several miRNAs, including *miR-135a,* might be involved in the process of atrial fibrillation. The physiological roles of these differentially expressed ncRNAs might be associated with the pathogenesis of atrial fibrillation, which might provide a new therapeutic target and/or a prognosis marker for patients with atrial fibrillation [14].

Atrial fibrillation is the most prevalent type of arrhythmia worldwide [16]. Atrial fibrosis is a hallmark feature of atrial structural remodeling in atrial fibrillation, which is regulated by the TGF-*β*1/Smad3 pathway. The dysregulation of tissue blockers of matrix metalloproteinases, including MMPs and TIMPs, linked to collagen upregulation is also a key factor in the development of atrial fibrillation. The *miR-146b-5p* might be an inhibitor of TIMP protein expression, which may be increased in atrial tissue with atrial fibrillation. Furthermore, some profibrotic markers, such as MMP9, TGFβ1, and COL1A1, have been shown to be downregulated by *miR-146b-5p* knockdown, while TIMP4 might trigger inverse expression patterns [17]. Another mechanism triggering atrial fibrillation is ferroptosis, which may be classified as iron-dependent cell death due to the excessive accumulation of peroxidized polyunsaturated fatty acids. Interestingly, the occurrence of ferroptosis in atrial fibrillation has been shown to be promoted by *miR-23a-3p*. In addition, *SLC7A11* is a direct target of *miR-23a-3p*, which might also be associated with ferroptosis [18]. Alternative miR-dependent gene regulation could elicit the initiation and/or progression of atrial fibrillation. For example, *miR-34a* could upregulate the expression of a tandem of P domains in a friable inner repairing K+ channel-linked acid-sensitive K+ channel 1 (TASK-1), which might promote a reduction in the resting membrane potential [19]. In this way, several ncRNAs might be involved in the regulation of the atrial expression of a potassium channel.

## 3. Bradycardia and Tachycardia

Bradycardia might be a heart dysfunction described by an extremely decreased heart rate, typically lower than 50 beats/min. Key etiological factors of bradycardia may include the dysregulation of sinus, atrial, or junctional bradycardia and/or an intricate transmission system with atrioventricular block. The most frequent form of bradycardia occurs asymptomatically during sleep and/or in athletes [20]. The only treatment for insistent bradycardia may be the placement of an everlasting pacemaker. Bradycardia is usually regulated by a range of molecules, including ncRNAs, at molecular levels. Transcription factors might also control the expression of genes involved in several bradycardias by modulating a variety of effectors, including miRNAs. For example, it has been demonstrated that the function of *miR-370-3p* could contribute to the development of bradycardia [21], which might be triggered via the reduced function of sinus node pacemaker channels with a related reduction of the ionic current. The *miR-370-3p* could directly bind to the mRNA of HCN4 to suppress its activity [21]. Similarly, *miR-486-3p* could suppress HCN4, inducing sinus node dysregulation such as bradycardia [22,23]. Supraventricular arrhythmia may be an example of tachycardia that begins in the upper compartments of the heart. Tachycardia is known to be considered a fast heartbeat, which might also be regulated by a range of molecules, including ncRNAs, at molecular levels. For example, the expression of *miR-1183* may be considerably upregulated in tachycardia. Raised expression of *miR-1183* may serve as a tissue biomarker for atrial remodeling, which might have a probable impact on cardiac disease [24]. Similarly, transcriptomic information even from the urine sample has revealed considerably decreased levels of miRNAs, including *miR-423*, *miR-1180*, *miR-3162*, *miR-3197*, *miR-3613*, *miR-6511*, and *miR-6763.* The changes in these miRNAs have been shown to be associated with increased expression of profibrotic markers such as Col I, Col III, fibronectin, and TGF-β. In particular, *miR-423* has been shown specifically to regulate calcium-control proteins, such as the phosphorylated and/or activated calmodulin-dependent protein kinase II, which might regulate the development of tachycardia [25]. In addition, the detection of miR-423 in human urine may be a potential novel approach for the diagnosis of atrial fibrillation [25].

## 4. Other Cardiac Rhythm Disorders

Ventricular arrhythmias have been described as irregular heartbeats that may derive from the ventricles. The origin of ventricular arrhythmias might be sympathetic remodeling that originates in myocardial infarction. Various ncRNAs are important regulators of inflammation and/or sympathetic remodeling following myocardial infarction. Several lncRNAs are potentially implicated in ventricular arrhythmias following myocardial infarction. For example, the lncRNA *LOC100911717 (LOC10)* might be increased in cardiac cells and macrophages in the infarcted heart area [26]. Another important factor in ventricular arrhythmias may be NLRP3 inflammasomes. It has been demonstrated that the role of *SOX2-OT* lncRNA in regulating NLRP3 inflammasome-mediated ventricular arrhythmias. The levels of *SOX2-OT* and NLRP3 inflammasomes could be intensified after ventricular arrhythmias [27]. Similarly, *miR-2355-3p* and *miR-1231* may play key roles in the regulation of ventricular arrhythmias [28,29], which might provide a potential diagnostic marker as well as a potential target for various treatments [28].

Long QT and short QT syndromes are known as the entities of Brugada syndrome [30]. Risk stratification for sudden cardiac death in patients with Brugada syndrome remains a major challenge. Brugada patients display a distinct miRNA expression profile compared with unaffected control individuals. For example, the *miR-145-5p* and *miR-585-3p* are associated with the standing position of Brugada patients, which might be the potential utility of leucocyte-derived miRNAs as prognostic biomarkers for Brugada syndrome [31,32]. Arrhythmogenic cardiomyopathy is a genetically determined disorder caused by mutations in proteins constituting desmosomes, which might be an inherited cardiomyopathy histologically categorized by the replacement of myocardium with inflammation, fibrofatty infiltration, and cardiomyocyte loss [33]. Inflammation might also be involved in the pathogenesis of arrhythmogenic cardiomyopathy with various inflammatory cytokines [34]. Consistently, it has been suggested that circulating levels of cytokine receptors (IL1-R1, IL6-R, and TNF-R1/R2) may be elevated and associated with the risk of arrhythmias in arrhythmogenic cardiomyopathy [35,36,37]. In particular, myocardial expression of IL-17 and/or TNF is substantially enhanced in all cases of arrhythmogenic cardiomyopathy [38]. In a large cohort of samples, the *miR-122-5p*, *miR-133b*, *miR-133a-3p*, *miR-142-3p*, *miR-183-5p,* and *miR-182-5p* could possess high diagnostic potential in arrhythmogenic cardiomyopathy patients [39,40]. The implication of miRNAs in the pathogenesis of arrhythmogenic cardiomyopathy has been the objective of multiple studies.

## 5. Gut Microbiota and Cardiac Arrhythmia

Disruption of the gut microbiota could induce cardiorespiratory morbidity. Consistently, modulation of autonomic homeostasis via the gut microbiota-brain axis could control the heart rate, independent of carotid body plasticity [41]. Sympathetic neuronal communication between the hypothalamic paraventricular nucleus and the gut might also be involved in the regulation of blood pressure [41]. The adult gut system might be an extremely diverse and dynamic ecosystem [42,43], which might employ specific physiological functions including immune regulation, gut mucosal protection, and conservation of nutritional metabolisms, in addition to the development of several diseases [44]. Some elements of the gut microbiota could stimulate the afferent fiber of the vagus nerve through the gut endocrine cells, which could also inspire the central nervous system (CNS). Depending on the material, the different SCFAs produced by the gut microbiota could activate the vagal afferent fibers in several ways. For example, butyric acid, a short fatty acid, directly affects afferent terminals [45,46]. An increase in the concentration of butyric acid in the colon may produce a significant hypotensive effect, which depends on afferent colonic vagus nerve signaling and GPR41/43 receptors [46]. Several investigations have considered the effect of food components on gut microbiota, which could be an imperative target for the forthcoming treatment of cardiac arrhythmias through the alteration of gut microbiota. For example, patients with arrhythmias may prefer to obtain more energy from animal fat [47]. Therefore, patients with cardiac arrhythmias may be frequently diagnosed with atherosclerosis. In addition, diabetes may also be a frequent co-morbidity in individuals with arrhythmias. Furthermore, the interaction between some drugs and their impact on the gut microbiota may result in uncontrolled arrhythmias [48,49,50]. Recent increasing data suggest that regulatory non-coding RNAs such as miRNAs, circular RNAs, and lncRNAs may affect host-microbe interactions. These ncRNAs have also been suggested as potential biomarkers in microbiome-associated disorders with a direct cross-talk between microbiome composition and ncRNAs [51]. Therefore, gut microbiota could affect responses to stimuli by host cells with modifications to their epigenome and/or gene expression. Recent data suggest that regulatory ncRNAs such as miRNAs, circular RNAs, and lncRNAs might affect host-microbe interactions. For example, miR-155 has been associated with inflammatory bowel diseases and might also be involved in cardiac remodeling following acute myocardial infarction [51]. Epigenetically, *miR-23a-3p* could lead to a Th17/Treg imbalance and participate in the progression of Graves’ disease [52], which might also be involved in the mechanism underlying atrial fibrillation. Consequently, gut microbiota could play a vital role in the pathogenesis of both atrial fibrillation and Graves’ disease [52]. The imbalance of Th17/Treg cells induced by the alteration of gut microbiota could also play a dynamic role in the pathogenesis of arrhythmias [53,54], suggesting that the immune response and arrhythmias are closely related. In particular, Th17 cells might be involved in the pathogenesis of atrial fibrillation, since increased levels of Th17-associated cytokines may be independently associated with an increased risk of atrial fibrillation [53]. The balance between Th17 and Treg cells has been deliberated to represent a paradigm for several inflammatory and auto-immune diseases. The inflammatory response might play a key role in the pathogenesis of atrial fibrillation, and the restoration of the Th17/Treg balance might represent a promising therapeutic target for treatment [54]. Interestingly, altered composition of the gut microbiota could change the expression of hepatic *miR-34a* [55], which might also be involved in the development of atrial fibrillation. In addition, there is a negative correlation between *miR-122-5p* and intestinal bacteria, including *Bacteriodes. uniformis* and *Phascolarctobacterium. Faecium* has been shown [56], suggesting that the crosstalk between miRNA and certain gut microbiota could regulate intracellular signal transduction by controlling the expression of key genes related to the development of arrhythmias [56]. As mentioned before, *miR-122-5p* could possess great diagnostic potential in arrhythmogenic cardiomyopathy patients [40,57] (Figure 2).

In these ways, data from several studies in both experimental animals and clinical humans may suggest that the role of the gut microbiota and its metabolites in arrhythmogenicity is well interconnected. Through a variety of pathways, the gut microbiota could significantly regulate the development of cardiac arrhythmias, which may provide an extensive range of possible therapeutic approaches [58,59]. For example, experimental evidence has indicated a bidirectional relationship between alterations in the categories of bacteria present in the gut and neurogenic diseases, including hypertension, which might also function as a molecular dispatch that could convert gut bacterial signals into a neural yield that could also influence hypertension and/or cardiac electrophysiology [58]. In addition, it has been suggested that the role of the gut microbiota and its metabolites might be involved in the pathogenesis as well as the prognostic value of many cardiovascular diseases [59]. Linking the possible interfering role of gut microbiota in the pharmacokinetics of medications used in the treatment of cardiac arrhythmias would also help to better understand the potential of certain probiotics [60,61].

## 6. Immune Pathway and Cardiac Arrhythmias

There is accumulative attention on mechanisms molecularly directing the onset and/or progression of cardiac arrhythmias due to the intricate interplay that could elicit several immune cells, enhancing atrial fibrosis [62]. The dysregulation of the immune pathway might meaningfully contribute to ion channel dysfunction and the initiation of cardiac arrhythmias [63]. In other words, the potential molecular mechanism of cardiac arrhythmias may involve inflammatory and immune pathways that cause ion channel dysfunction, in which a variety of signaling systems might be involved [63]. Consecutively, the interaction between immune cells and atrial myocytes might augment atrial electrical and/or practical remodeling, which might be responsible for the progression of cardiac arrhythmias [64]. Interestingly, some ncRNAs are involved in the ribonucleolytic cleavage of the target mRNA [65], and several members of the APRO family have been revealed to be associated with the cytoplasmic mRNA deadenylation [66,67]. Similarly, several APRO family proteins could cooperate with the poly (A) ribonuclease complex [68,69]. These APRO family proteins (Btg1, Btg2, Tob1, and Tob2) might be key modulators of microRNAs [70], which may be extremely involved in the regulation of immune cells [71]. Therefore, APRO family proteins have been shown to be involved in immune-related disorders, including ulcerative colitis and cancer [72,73,74]. Interestingly, it has been shown that one of the APRO family members, BTG1, may be involved in the regulation of ion channel-related gene expression [75,76]. Remarkably, it has been shown that BTG1 could promote the deadenylation and/or degradation of mRNA to secure T cell quiescence [77], suggesting that BTG1 might be involved in a key mechanism underlying T cell quiescence. In addition, another APRO family member, Tob1, might be a key regulatory molecule in the process of endothelialization in the repair of atrial septal defects [78]. Tob1 might also play a fundamental role in keeping cells in a quiescent state, thereby obstructing the cellular proliferation of normal and/or cancer cells [79]. Tob1 may be inactivated in the cells of many human cancers. Therefore, Tob1 has been considered a tumor-suppressor protein [80].

## 7. Future Perspectives

The gut microbiota might play a significant role in human health by modulating the function of the immune system [81]. For example, the immune-regulating potential of gut microbiota could mediate some anti-cancer therapeutic functions, specifically targeting anti-CTLA-4 and anti-PD-L1 treatments that could show better clinical outcomes in patients with some species of Bacteroides and Bifidobacterium, respectively [81,82]. However, there is also much information saying that the gut microbiome has a noteworthy role in reducing the effectiveness of immunotherapy, which could lead to crucial resistance to immune checkpoint therapy [81]. A beneficial response might be observed in the existence of certain gut microbes, whereas microbial imbalance (dysbiosis) due to antibiotic treatment may be associated with poor response to anti-CTLA-4, anti-PD-1, and/or anti-PD-L1 immunotherapies [81]. It might be encouraging to use specific bacteria and their metabolites to restore the beneficial microbiome. Greater benefits of cancer therapies with anti-PD-1, anti-PD-L1, or anti-CTLA-4 monoclonal antibodies could be potentially detected in cancer patients supplemented with some bacterial species, including *Enterococcus hirae*, *Akkermansia mucinifila*, and/or *Faecalibacterium prausnitzii* [82]. This phenomenon has been hypothesized to be related to the action of gut microbiota just on dendritic cells, which are essential to trigger suitable T-cell responses [83]. Additionally, short-chain fatty acids (SCFAs) formed within the gut microbiota might also modulate those anti-CTLA-4 and anti-PD-1 immune responses as well as their anti-tumor efficacy [84]. SCFAs are categorized by having fewer than six carbons, such as acetate, propionate, and butyrate, which are mostly produced in the gut as fermentation products. Therefore, the human gut microbiota or their various metabolites, including SCFAs, have become a possible therapeutic target for the development of cancer interventions as well as for the prevention of cardio-metabolic disorders [85]. Interestingly, there are remarkable associations between gut microbiota and the microglia in the brain, which might suggest the profound crosstalk of gut microbiota and nerve function via the important roles of microglia [86]. Therefore, some modifications in the composition of the gut microbiota might play a critical role in the pathogenesis of several diseases. Remarkably, it has been presumed that cardiovascular diseases, including cardiac arrhythmia, might link to the pathogenesis that is based on the engram memory system [87,88,89] (Figure 3). The engram memory system in the brain could remember the information of a definite inflammation in the body that might be involved in the pathogenesis of various diseases, including ulcerative colitis and/or cardiovascular diseases, in which the immunity-linked processes might be associated with the neuronal responses to memory engrams [90]. Future work should define at the molecular level how this pathway could precisely interfere with regulating cardiac arrhythmias. In particular, upcoming research should focus on the identification of disease-specific engram preservation over time during the latent period of cardiovascular diseases.

## 8. Conclusions

Several ncRNAs could be involved in the development of certain cardiac arrhythmias, in which the association between gut and heart might play an important role. In addition, the correlation between gut and brain, as well as the correlation between brain and immunity, might also influence the development of various cardiovascular diseases, including cardiac arrhythmias. An in-depth knowledge of the role of ncRNAs and/or gut microbiota in cardiac arrhythmias might be valuable for developing innovative clinical diagnosis and treatment.

## Figures and Tables

**Figure 1 genes-14-01736-f001:**
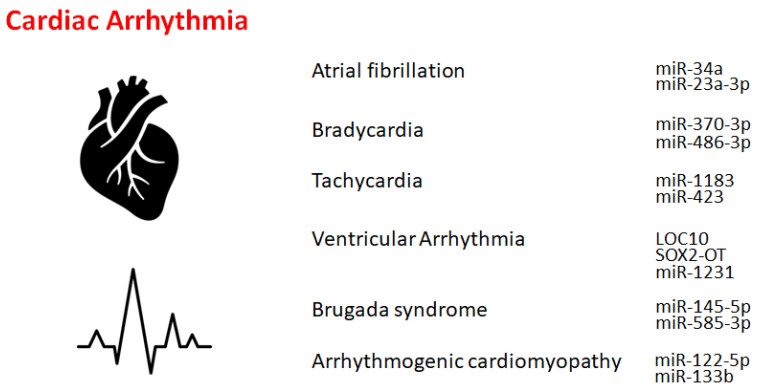
Schematic representation of the players of ncRNAs involved in cardiac arrhythmias. Example ncRNA molecules are shown for each cardiac arrhythmia.

**Figure 2 genes-14-01736-f002:**
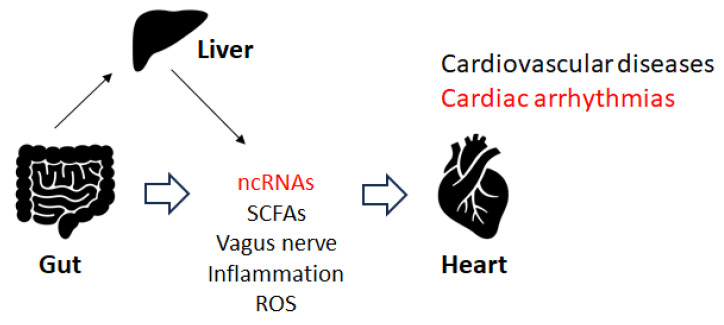
Several factors and/or inflammation with ROS may affect the development of cardiovascular diseases, including cardiac arrhythmias. Note that some critical pathways have been omitted for clarity.

**Figure 3 genes-14-01736-f003:**
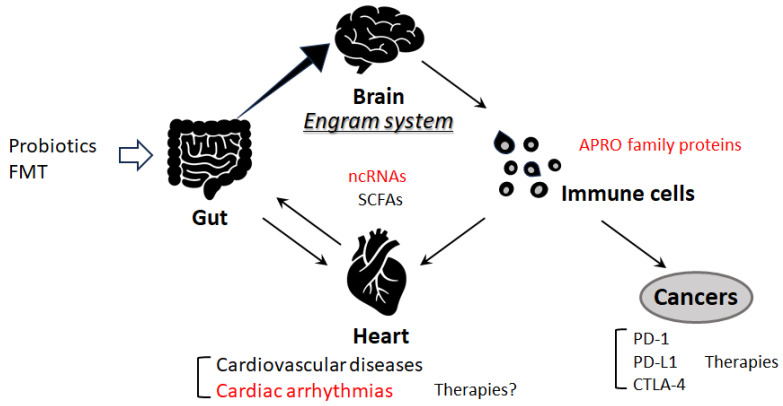
Schematic representation of the potential inhibitory tactics against the pathogenesis of cardiovascular diseases, including cardiac arrhythmias. Some kinds of probiotics and/or fecal microbiota transplantation (FMT) could contribute to the alteration of the gut microbial community through the alteration of ncRNA and/or SCFA production, which could be beneficial for the treatment of cardiovascular diseases. Note that several important activities, such as inflammatory reactions, autophagy initiation, and ROS production, have been omitted for clarity.

## Data Availability

Not applicable.

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
