# Peer review of "Non-Coding RNAs and Gut Microbiota in the Pathogenesis of Cardiac Arrhythmias: The Latest Update"

_genes, 2023, doi:10.3390/genes14091736_

Round 1
Reviewer 1 Report
Naoko Suga and colleagues wrote a review article entitled "Non-coding RNAs and gut microbiota might be involved in the pathogenesis of cardiac arrhythmias", in which a systematic description of the role of non-coding RNAs in the pathogenesis of different cardiac arrhythmias is provided. The authors conclude that "An in-depth knowledge of the role of ncRNAs in cardiac arrhythmias may be valuable for progressing new clinical diagnosis and treatment."
Although the topic is of potential interest for both the cardiology and the basic science audience, according to this reviewer, several major limitations offset the value of this work.
1) First and foremost, the description of action potential's physiological mechanisms, of pathophysiology leading to cardiac arrhythmias, and of arrhythmias themselves is too naive, and the presented concepts are too generic, limiting the utility of this paper.
2) The english style is inadequate. For example, in the introduction, "heterogenic" is used instead of "heterogeneous"; furthermore "a comprehensive arrhythmogenicity study have been briefly summarized" is stated, which is hard to understand, and so on. Many short phrases are used, which makes the text difficult to follow.
Therefore, extensive work is necessary before this paper might become acceptable for publication in the prestigious Genes journal.
The quality of english is scarce, with several mistakes in terms and verb conjugation.
Author Response
For Reviewer1
Naoko Suga and colleagues wrote a review article entitled "Non-coding RNAs and gut microbiota might be involved in the pathogenesis of cardiac arrhythmias", in which a systematic description of the role of non-coding RNAs in the pathogenesis of different cardiac arrhythmias is provided. The authors conclude that "An in-depth knowledge of the role of ncRNAs in cardiac arrhythmias may be valuable for progressing new clinical diagnosis and treatment." Although the topic is of potential interest for both the cardiology and the basic science audience, according to this reviewer, several major limitations offset the value of this work.
1) First and foremost, the description of action potential's physiological mechanisms, of pathophysiology leading to cardiac arrhythmias, and of arrhythmias themselves is too naive, and the presented concepts are too generic, limiting the utility of this paper.
Exactly. The aim of this review article may be to connect the pathophysiology of cardiac arrhythmias to genetic factors and/or gut microbiota, but not to anything else such as electronics or neurology. As you know, there is scarce evidence for the connection at present. Therefore, the aim of this review article may be also make some researchers encourage to perform investigations to get such valuable evidences for the development of clinical application utilizing this concept.
2) The english style is inadequate. For example, in the introduction, "heterogenic" is used instead of "heterogeneous"; furthermore "a comprehensive arrhythmogenicity study have been briefly summarized" is stated, which is hard to understand, and so on.
“Heterogenous” has been replaced with the “heterogenic” in the introduction. In addition, the sentence has been amended to “comprehensive studies related to the arrhythmogenicity of ncRNAs have been briefly summarized” for the easy readability.
Many short phrases are used, which makes the text difficult to follow. Therefore, extensive work is necessary before this paper might become acceptable for publication in the prestigious Genes journal. The quality of english is scarce, with several mistakes in terms and verb conjugation.
According to these suggestions, we have gone over the text/abstract and amended typos and grammatical errors as much as possible to improve the manuscript more helpful to the readers.
Reviewer 2 Report
Suga et al attempt to review the research on the NC-RNA involved in gut microbiota and arrhythmiogenesis
The structure, language are fine. Authors should rephrase the goal of this study in the end of the introduction. It is an overview of the relevant literature.
Since authors decided to include gut microbiota, they should be take into consideration the intestine autonomic nervous system
as well. doi: 10.1007/978-3-319-56246-9_20, https://www.ncbi.nlm.nih.gov/pmc/articles/PMC8037288/
Author Response
For Reviewer2
Suga et al attempt to review the research on the NC-RNA involved in gut microbiota and arrhythmiogenesis. The structure, language are fine. Authors should rephrase the goal of this study in the end of the introduction. It is an overview of the relevant literature.
Thank you so much for the good evaluation to our manuscript.
Since authors decided to include gut microbiota, they should be take into consideration the intestine autonomic nervous system as well. doi: 10.1007/978-3-319-56246-9_20, https://www.ncbi.nlm.nih.gov/pmc/articles/PMC8037288/
Thank you again for the good suggestion. Accordingly, we have considerably amended the manuscript with further taking into the deliberation of intestine autonomic nervous system.
Reviewer 3 Report
Dear Editor,
I have read with extreme interest the paper by Dr Suga et al. entitled “
Non-coding RNAs and gut microbiota might be involved in the 2 pathogenesis of cardiac arrhythmias”.
Both arguments (gut microbioma and ncRNAs) are of paramount importance in translational research for the clinical implications the imbalances and the related disorders of these two biological entities.
While for the ncRNA and cardiovascular diseases (CVDs) disorders there is a plethora of causal associations with molecular therapies against this class of RNAs, for the gut microbioma we are still at the beginning. Indeed, fecal microbial community changes (in terms of species and their relative abundance) are associated with numerous disease states, including CVD. However, such data are merely associative. A causal contribution for gut microbiota in CVD has been further supported by a multitude of more direct experimental evidence. Indeed, gut microbiota transplantation studies, specific gut microbiota-dependent pathways, and downstream metabolites have all been shown to influence host metabolism and CVD, sometimes through specific identified host receptors. Multiple metaorganismal pathways (involving both microbe and host) both impact CVD in animal models and show striking clinical associations in human studies.
One of the weak point of the present paper is the lack of description of changes in terms species (bacterial and/or fungi) and the potential impact of the immune system and therefore the CV apparatus.
The ncRNA part of the paper is weel described with convincing association studies.
I would reinforce the microbioma section providing actual examples.
Author Response
For Reviewer3
I have read with extreme interest the paper by Dr Suga et al. entitled “Non-coding RNAs and gut microbiota might be involved in the 2 pathogenesis of cardiac arrhythmias”. Both arguments (gut microbioma and ncRNAs) are of paramount importance in translational research for the clinical implications the imbalances and the related disorders of these two biological entities.
While for the ncRNA and cardiovascular diseases (CVDs) disorders there is a plethora of causal associations with molecular therapies against this class of RNAs, for the gut microbioma we are still at the beginning. Indeed, fecal microbial community changes (in terms of species and their relative abundance) are associated with numerous disease states, including CVD. However, such data are merely associative. A causal contribution for gut microbiota in CVD has been further supported by a multitude of more direct experimental evidence. Indeed, gut microbiota transplantation studies, specific gut microbiota-dependent pathways, and downstream metabolites have all been shown to influence host metabolism and CVD, sometimes through specific identified host receptors. Multiple metaorganismal pathways (involving both microbe and host) both impact CVD in animal models and show striking clinical associations in human studies.
Thank you so much for the good evaluation to our manuscript. We think so, too.
One of the weak point of the present paper is the lack of description of changes in terms species (bacterial and/or fungi) and the potential impact of the immune system and therefore the CV apparatus. The ncRNA part of the paper is weel described with convincing association studies. I would reinforce the microbioma section providing actual examples.
Exactly, it is correct. We have added some description of terms for the related species. However, there is scarce evidence showing actual examples including species term and/or efficacy, at present. The aim of this paper would make some researchers encourage to perform basic or clinical investigations for getting actual evidence, near future.
Reviewer 4 Report
This review is an interesting summarization of the current knowledge about the role of non-coding RNAs (ncRNAs) in the pathogenesis of cardiac arrhythmia, trying to enlighten the molecular mechanism beneath cardiac arrhythmia. The authors did a great job to offer the theoretical basis of ncRNAs in the most common arrhythmic diseases including atrial fibrillation and bradycardia. I appreciate that the authors took the effort to explore the role of gut microbes in the ncRNAs- arrhythmia axis. Overall, the manuscript and figures are clearly organized and provide updated information. I listed several concerns that need to be addressed.
1. The most attractive part of this review is the role of gut microbes in the ncRNAs-arrhythmia axis, which, however, is less informative. Only a smaller effort was taken to enlighten this topic in the late half of this manuscript. I encourage the authors to enroll more research on this interesting topic. Less speculation, more evidence.
2. It is suggested to avoid “might” in the title of this manuscript even though the authors are trying to be rigorous. Because the readers won’t appreciate an unsettled story that is even doubted by the authors themselves. Consider “Non-coding RNAs, and gut microbiota, in the pathogenesis of cardiac arrhythmias: the latest updating”.
Author Response
For Reviewer4
This review is an interesting summarization of the current knowledge about the role of non-coding RNAs (ncRNAs) in the pathogenesis of cardiac arrhythmia, trying to enlighten the molecular mechanism beneath cardiac arrhythmia. The authors did a great job to offer the theoretical basis of ncRNAs in the most common arrhythmic diseases including atrial fibrillation and bradycardia. I appreciate that the authors took the effort to explore the role of gut microbes in the ncRNAs- arrhythmia axis. Overall, the manuscript and figures are clearly organized and provide updated information.
Thank you so much for the good evaluation to our manuscript.
I listed several concerns that need to be addressed.
- The most attractive part of this review is the role of gut microbes in the ncRNAs-arrhythmia axis, which, however, is less informative. Only a smaller effort was taken to enlighten this topic in the late half of this manuscript. I encourage the authors to enroll more research on this interesting topic. Less speculation, more evidence.
Correct. However, there is very limited evidence showing actual examples, at present. The aim of this paper would make several researchers encourage to achieve basic and/or clinical investigations for getting definite evidence, near future.
- It is suggested to avoid “might” in the title of this manuscript even though the authors are trying to be rigorous. Because the readers won’t appreciate an unsettled story that is even doubted by the authors themselves. Consider “Non-coding RNAs, and gut microbiota, in the pathogenesis of cardiac arrhythmias: the latest updating”.
This is very good suggestion to us. The title has been replaced with the above, accordingly. Thank you so much.
Round 2
Reviewer 4 Report
This review is an interesting summarization of the current knowledge on the role of non-coding RNAs (ncRNAs) in the pathogenesis of cardiac arrhythmia, trying to enlighten the molecular mechanism beneath cardiac arrhythmia. The authors did a great job of offering the theoretical basis of ncRNAs in the most common arrhythmic diseases including atrial fibrillation and bradycardia. I appreciate that the authors took the effort to explore the role of gut microbes in the ncRNAs- arrhythmia axis. Overall, the manuscript and figures are clearly organized and provide updated information. The authors responded sufficiently to my questions and made detailed revisions. After explanations, the conclusion is less vulnerable.